# Implicit bias of gradient descent for mean squared error regression with wide neural networks

## Abstract

We investigate gradient descent training of wide neural networks and the corresponding implicit bias in function space. For 1D regression, we show that the solution of training a width-$n$ shallow ReLU network is within $n^{-1/2}$ of the function which fits the training data and whose difference from initialization has smallest 2-norm of the weighted second derivative with respect to the input. The curvature penalty function $1/\zeta$ is expressed in terms of the probability distribution that is utilized to initialize the network parameters, and we compute it explicitly for various common initialization procedures. For instance, asymmetric initialization with a uniform distribution yields a constant curvature penalty, and thence the solution function is the natural cubic spline interpolation of the training data. While similar results have been obtained in previous works, our analysis clarifies important details and allows us to obtain significant generalizations. In particular, the result generalizes to multivariate regression and different activation functions. Moreover, we show that the training trajectories are captured by trajectories of spatially adaptive smoothing splines with decreasing regularization strength.

*Keywords.* Implicit bias, overparametrized neural network, cubic spline interpolation, spatially adaptive smoothing spline, effective capacity.

## 1 Introduction

Understanding why neural networks trained in the overparametrized regime and without explicit regularization generalize well in practice is an important problem (Zhang et al., 2017). Some form of capacity control different from network size must be at play (Neyshabur et al., 2014) and specifically the implicit bias of parameter optimization has been identified to play a key role (Neyshabur et al., 2017). By implicit bias we mean that among the many hypotheses that fit the training data, the algorithm selects one which satisfies additional properties that may be beneficial for its performance on new data.

Jacot et al. (2018) and Lee et al. (2019) showed that the training dynamics of shallow and deep wide neural networks is well approximated by that of the linear Taylor approximation of the models at a suitable initialization. Chizat et al. (2019) observe that a model can converge to zero training loss while hardly varying its parameters, a phenomenon that can be attributed to scaling of the output weights and makes the model behave as its linearization around the initialization. Zhang et al. (2019) consider linearized models for regression problems and show that gradient flow finds the global minimum of the loss function which is closest to initialization in parameter space. This type of analysis connects with trajectory based analysis of neural networks (Saxe et al., 2014). Oymak and Soltanolkotabi (2019) studied the overparametrized neural networks directly and showed that gradient descent finds a global minimizer of the loss function which is close to the initialization. Towards interpreting parameters in function space, Savarese et al. (2019) and Ongie et al. (2020) studied infinite-width neural networks with parameters having bounded norm, in 1D and multi-dimensional input spaces, respectively. They showed that, under a standard parametrization, the complexity of the functions represented by the network, as measured by the 1-norm of the second derivative, can be controlled by the 2-norm of the parameters. Using these results, one can show that gradient descent with $\ell_2$ weight penalty leads to simple functions. Sahs et al. (2020) relates function properties, such as breakpoint and slope distributions, to the distributions of the network parameters.

The implicit bias of parameter optimization has been investigated in terms of the properties of the loss function at the points reached by different optimization methodologies (Keskar et al., 2017; Wu et al., 2017; Dinh et al., 2017). In terms of the solutions, Maennel et al. (2018) show that gradient flow for shallow networks with rectified linear units (ReLU) initialized close to zero quantizes features in a way that depends on the training data but not on the network size. Williams et al. (2019) obtained results for 1D regression contrasting the kernel and adaptive regimes. Soudry et al. (2018) show that in classification problems with separable data, gradient descent with linear networks converges to a max-margin solution. Gunasekar et al. (2018b) present a result on implicit bias for deep linear convolutional networks, and Ji and Telgarsky (2019) study non-separable data. Chizat and Bach (2020) show that gradient flow for logistic regression with infinitely wide two-layer networks yields a max-margin classifier in a certain space. Gunasekar et al. (2018a) analyze the implicit bias of different optimization methods (natural gradient, steepest and mirror descent) for linear regression and separable linear classification problems, and obtain characterizations in terms of minimum norm or max-margin solutions.

In this work, we study the implicit bias of gradient descent for regression problems. We focus on wide ReLU networks and describe the bias in function space. In Section 2 we provide settings and notation. We present our main results in Section 3, and develop the main theory in Sections 4 and 5. In the interest of a concise presentation, technical proofs and extended discussions are deferred to appendices.

## 2 NOTATION AND PROBLEM SETUP

Consider a fully connected network with $d$ inputs, one hidden layer of width $n$, and a single output. For any given input $x \in \mathbb{R}^d$, the output of the network is

$$f(x,\theta) = \sum_{i=1}^{n} W_i^{(2)} \phi(\langle W_i^{(1)}, x \rangle + b_i^{(1)}) + b^{(2)}, \tag{1}$$

where $\phi$ is a point-wise activation function, $W^{(1)} \in \mathbb{R}^{n \times d}$, $W^{(2)} \in \mathbb{R}^n$, $b^{(1)} \in \mathbb{R}^n$ and $b^{(2)} \in \mathbb{R}$ are the weights and biases of layer $l = 1,2$. We write $\theta = \text{vec}(\cup_{l=1}^2 \{W^{(l)}, b^{(l)}\})$ for the vector of all network parameters. These parameters are initialized by independent samples of pre-specified random variables $\mathcal{W}$ and $\mathcal{B}$ in the following way:

$$\begin{aligned} W_{i,j}^{(1)} &\overset{d}{=} \sqrt{1/d}\, \mathcal{W}, \quad b_i^{(1)} \overset{d}{=} \sqrt{1/d}\, \mathcal{B} \\ W_i^{(2)} &\overset{d}{=} \sqrt{1/n}\, \mathcal{W}, \quad b^{(2)} \overset{d}{=} \sqrt{1/n}\, \mathcal{B}. \end{aligned} \tag{2}$$

More generally, we will also allow weight-bias pairs to be sampled from a joint distribution of $(\mathcal{W}, \mathcal{B})$ which we only assume to be sub-Gaussian. In the analysis of Jacot et al. (2018); Lee et al. (2019), $\mathcal{W}$ and $\mathcal{B}$ are Gaussian $\mathcal{N}(0, \sigma^2)$. In the default initialization of PyTorch, $\mathcal{W}$ and $\mathcal{B}$ have uniform distribution $\mathcal{U}(-\sigma, \sigma)$. The setting (1) is known as the standard parametrization. Some works (Jacot et al., 2018; Lee et al., 2019) utilize the so-called NTK parametrization, where the factor $\sqrt{1/n}$ is carried outside of the trainable parameter. If we fix the learning rate for all parameters, gradient descent leads to different trajectories under these two parametrizations. Our results are presented for the standard parametrization. Details on this in Appendix C.3.

We consider a regression problem for data $\{(x_j, y_j)\}_{j=1}^M$ with inputs $\mathcal{X} = \{x_j\}_{j=1}^M$ and outputs $\mathcal{Y} = \{y_j\}_{j=1}^M$. For a loss function $\ell : \mathbb{R} \times \mathbb{R} \to \mathbb{R}$, the empirical risk of our function is $L(\theta) = \sum_{j=1}^M \ell(f(x_j, \theta), y_j)$. We use full batch gradient descent with a fixed learning rate $\eta$ to minimize $L(\theta)$. Writing $\theta_t$ for the parameter at time $t$, and $\theta_0$ for the initialization, this defines an iteration

$$\theta_{t+1} = \theta_t - \eta \nabla L(\theta) = \theta_t - \eta \nabla_\theta f(\mathcal{X}, \theta_t)^T \nabla_{f(\mathcal{X}, \theta_t)} L, \tag{3}$$

where $f(\mathcal{X}, \theta_t) = [f(x_1, \theta_t), ..., f(x_M, \theta_t)]^T$ is the vector of network outputs for all training inputs, and $\nabla_{f(\mathcal{X}, \theta_t)} L$ is the gradient of the loss with respect to the model outputs. We will use subscript $i$ to index neurons and subscript $t$ to index time. Let $\hat{\Theta}_n$ be the empirical neural tangent kernel (NTK) of the standard parametrization at time 0, which is the matrix $\hat{\Theta}_n = \frac{1}{n} \nabla_\theta f(\mathcal{X}, \theta_0) \nabla_\theta f(\mathcal{X}, \theta_0)^T$.

## 3 MAIN RESULTS AND DISCUSSION

We obtain a description of the implicit bias in function space when applying gradient descent to regression problems with wide ReLU neural networks. We prove the following result in Appendix D. An interpretation of the result and generalizations are given further below.

**Theorem 1** (Implicit bias of gradient descent in wide ReLU networks). *Consider a feedforward network with a single input unit, a hidden layer of $n$ rectified linear units, and a single linear output unit. Assume standard parametrization* (1) *and that for each hidden unit the input weight and bias are initialized from a sub-Gaussian $(\mathcal{W}, \mathcal{B})$* (2) *with joint density $p_{\mathcal{W},\mathcal{B}}$. Then, for any finite data set $\{(x_j, y_j)\}_{j=1}^M$ and sufficiently large $n$ there exist constant $u$ and $v$ so that optimization of the mean square error on the adjusted training data $\{(x_j, y_j - ux_j - v)\}_{j=1}^M$ by full-batch gradient descent with sufficiently small step size converges to a parameter $\theta^*$ for which the output function $f(x, \theta^*)$* (1) *attains zero training error. Furthermore, letting $\zeta(x) = \int_{\mathbb{R}} |W|^3 p_{\mathcal{W},\mathcal{B}}(W, -Wx) \, \mathrm{d}W$ and $S = \mathrm{supp}(\zeta) \cap [\min_i x_j, \max_i x_j]$, we have $\|f(x, \theta^*) - g^*(x)\|_2 = O(n^{-\frac{1}{2}}), x \in S$ (the 2-norm over $S$) with high probability over the random initialization $\theta_0$, where $g^*$ solves following variational problem:*

$$\min_{g \in C^2(S)} \quad \int_S \frac{1}{\zeta(x)} (g''(x) - f''(x, \theta_0))^2 \, \mathrm{d}x \tag{4}$$
$$\text{subject to} \quad g(x_j) = y_j - ux_j - v, \quad j = 1, ..., M.$$

**Interpretation** An intuitive interpretation of the theorem is that at those regions of the input space where $\zeta$ is smaller, we can expect the difference between the functions after and before training to have a small curvature. We may call $\rho = 1/\zeta$ a curvature penalty function. The bias induced from initialization is expressed explicitly. We note that under suitable asymmetric parameter initialization (see Appendix C.2), it is possible to achieve $f(\cdot, \theta_0) \equiv 0$. Then the regularization is on the curvature of the output function itself. In Theorem 9 we obtain the explicit form of $\zeta$ for various common parameter initialization procedures. In particular, when the parameters are initialized independently from a uniform distribution on a finite interval, $\zeta$ is constant and the problem is solved by the natural cubic spline interpolation of the data. The adjustment of the training data simply accounts for the fact that second derivatives define a function only up to linear terms. In practice we can use the coefficients $a$ and $b$ of linear regression $y_j = ax_j + b + \epsilon_j$, $j = 1, ..., M$, and set the adjusted data as $\{(x_j, \epsilon_j)\}_{j=1}^M$. Although Theorem 1 describes the gradient descent training with the linearly adjusted data, this result can also approximately describe training with the original training data. Further details are provided in Appendix L.

We illustrate Theorem 1 numerically in Figure 1 and more extensively in Appendix A. In close agreement with the theory, the solution to the variational problem captures the solution of gradient descent training uniformly with error of order $n^{-1/2}$. To illustrate the effect of the curvature penalty function, Figure 1 also shows the solutions to the variational problem for different values of $\zeta$ corresponding to different initialization distributions. We see that at input points where $\zeta$ is small / peaks strongly, the solution function tends to have a lower curvature / be able to use a higher curvature in order to fit the data.

With the presented bias description we can formulate heuristics for parameter initialization either to ease optimization or also to induce specific smoothness priors on the solutions. In particular, by Proposition 8 any curvature penalty $1/\zeta$ can be implemented by an appropriate choice of the parameter initialization distribution. By our analysis, the effective capacity of the model, understood as the set of possible output functions after training, is adapted to the size $M$ of the training dataset and is well captured by a space of cubic splines relative to the initial function. This is a space with dimension of order $M$ independently of the number of parameters of the network.

**Strategy of the proof** In Section 4, we observe that for a linearized model gradient descent with sufficiently small step size finds the minimizer of the training objective which is closest to the initial parameter (similar to a result by Zhang et al., 2019). Then Theorem 4 shows that the training dynamics of the linearization of a wide network is well approximated in parameter and function space by that of a lower dimensional linear model which trains only the output weights. This property is sometimes taken for granted and we show that it holds for the standard parametrization, although it does not hold for the NTK parametrization (defined in Appendix C.3), which leads to the adaptive regime. In Section 5, for networks with a single input and a single layer of ReLUs, we relate the implicit bias of gradient descent in parameter space to an alternative optimization problem. In Theorem 5 we show that the solution of this problem has a well defined limit as the width of the network tends to infinity,

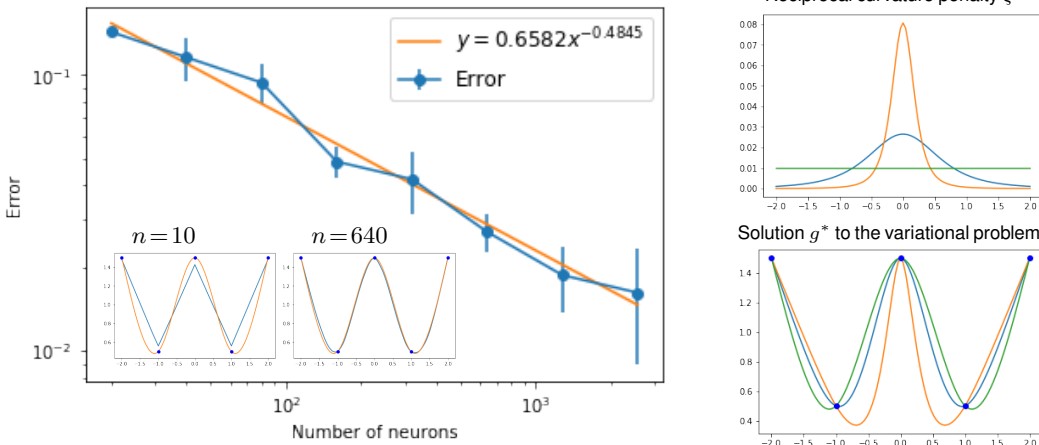

Figure 1: Illustration of Theorem 1. Left: Uniform error between the solution $g^*$ to the variational problem and the functions $f(\cdot,\theta^*)$ obtained by gradient descent training of a neural network (in this case with uniform initialization $W \sim U(-1,1)$, $B \sim U(-2,2)$), against the number of neurons. The inset shows examples of the trained networks (blue) alongside with the training data (dots) and the solution to the variational problem (orange). Right: Effect of the curvature penalty function on the shape of the solution function. The bottom shows $g^*$ for various different $\zeta$ shown at the top. Again dots are training data. The green curve is for $\zeta$ constant on $[-2,2]$, derived from initialization $\mathcal{W} \sim U(-1,1)$, $\mathcal{B} \sim U(-2,2)$; blue is for $\zeta(x) = 1/(1+x^2)^2$, derived from $\mathcal{W} \sim N(0,1)$, $\mathcal{B} \sim N(0,1)$; and orange for $\zeta(x) = 1/(0.1+x^2)^2$, derived from $\mathcal{W} \sim N(0,1)$, $\mathcal{B} \sim N(0,0.1)$. Theorem 9 shows how to compute $\zeta$ for the above distributions.

which allows us to obtain a variational formulation. In Theorem 6 we translate the description of the bias from parameter space to function space. In Theorem 9 we provide explicit descriptions of the weight function for various common initialization procedures. Finally, we can utilize recent results bounding the difference in function space of the solutions obtained from training a wide network and its linearization (Lee et al., 2019, Theorem H.1).

**Generalizations** Theorem 4 has several generalizations elaborated in Appendix P. For multivariate regression, we have the following theorem.

**Theorem 2** (Multivariate regression). *Use the same network setting as in Theorem 1 except that the number of input units changes to $d$. Assume that for each hidden unit the input weight and bias are initialized from a sub-Gaussian $(\mathcal{W},\mathcal{B})$ where $\mathcal{W}$ is a $d$-dimensional random vector and $\mathcal{B}$ is a random variable. Then, for any finite data set $\{(\mathbf{x}_j,y_j)\}_{i=1}^M$ and sufficiently large $n$ there exist constant vector $\mathbf{u}$ and constant $v$ so that optimization of the mean square error on the adjusted training data $\{(\mathbf{x}_j,y_j-\langle\mathbf{u},\mathbf{x}_j\rangle-v)\}_{j=1}^M$ by full-batch gradient descent with sufficiently small step size converges to a parameter $\theta^*$ for which $f(\mathbf{x},\theta^*)$ attains zero training error. Furthermore, let $\mathcal{U} = \|\mathcal{W}\|_2$, $\mathcal{V} = \mathcal{W}/\|\mathcal{W}\|_2$, $\mathcal{C} = -\mathcal{B}/\|\mathcal{W}\|_2$ and $\zeta(\mathbf{V},c) = p_{\mathcal{V},\mathcal{C}}(\mathbf{V},c)\mathbb{E}(\mathcal{U}^2|\mathcal{V} = \mathbf{V},\mathcal{C} = c)$ where $p_{\mathcal{V},\mathcal{C}}$ is the joint density of $(\mathcal{V},\mathcal{C})$. Then we have $\|f(\mathbf{x},\theta^*) - g^*(\mathbf{x})\|_2 = O(n^{-\frac{1}{2}})$, $\mathbf{x} \in \mathbb{R}^d$ (the 2-norm over $\mathbb{R}^d$) with high probability over the random initialization $\theta_0$, where $g^*$ solves following variational problem:*

$$\min_{g \in C(\mathbb{R}^d)} \quad \int_{\text{supp}(\zeta)} \frac{\left(\mathcal{R}\{(-\Delta)^{(d+1)/2}(g-f(\cdot,\theta_0))\}(\mathbf{V},c)\right)^2}{\zeta(\mathbf{V},c)} \, \mathrm{d}\mathbf{V}\mathrm{d}c$$

$$\text{subject to} \quad g(\mathbf{x}_j) = y_j, \quad j = 1,...,M, \tag{5}$$

$$\mathcal{R}\{(-\Delta)^{(d+1)/2}(g-f(\cdot,\theta_0))\}(\mathbf{V},c) = 0, \quad (\mathbf{V},c) \notin \text{supp}(\zeta).$$

*Here $\mathcal{R}$ is the Radon transform which is defined by $\mathcal{R}\{f\}(\boldsymbol{\omega},b) := \int_{\langle\boldsymbol{\omega},\mathbf{x}\rangle=b} f(\mathbf{x})\mathrm{d}s(\mathbf{x})$, and the power of the negative Laplacian $(-\Delta)^{(d+1)/2}$ is the operator defined in Fourier domain by $\widehat{(-\Delta)^{(d+1)/2}}f(\boldsymbol{\xi}) = \|\boldsymbol{\xi}\|^{d+1}\widehat{f}(\boldsymbol{\xi})$.*

For different activation functions, we have the following corollary.

**Corollary 3** (Different activation functions). *Use the same setting as in Theorem 1 except that we use the activation function $\phi$ instead of ReLU. Suppose that $\phi$ is a Green's function of a linear operator* L, *i.e.* $L\phi = \delta$, *where $\delta$ denotes the Dirac delta function. Assume that the activation function $\phi$ is homogeneous of degree $k$, i.e. $\phi(ax) = a^k\phi(x)$ for all $a > 0$. Then we can find a function $p$ satisfying* $Lp \equiv 0$ *and adjust training data* $\{(x_j, y_j)\}_{j=1}^M$ *to* $\{(x_j, y_j - p(x_j))\}_{j=1}^M$. *After that, the statement in Theorem 1 holds with the variational problem (4) changed to*

$$\min_{g \in C^2(S)} \int_S \frac{1}{\zeta(x)}[L(g(x) - f(x, \theta_0))]^2 \, dx \quad s.t. \quad g(x_j) = y_j - p(x_j), \quad j = 1, ..., M, \quad (6)$$

*where* $\zeta(x) = p_{\mathcal{C}}(x)\mathbb{E}(\mathcal{W}^{2k} | \mathcal{C} = x)$ *and* $S = \text{supp}(\zeta) \cap [\min_i x_i, \max_i x_i]$.

Moreover, our method allows us to describe the optimization trajectory in function space (see Appendix N). If we substitute constraints $g(x_j) = y_j$ in (4) by a quadratic term $\frac{1}{\lambda}\frac{1}{M}\sum_{j=1}^M (g(x_j) - y_j)^2$ added to the objective, we obtain the variational problem for a so-called spatially adaptive smoothing spline (see Abramovich and Steinberg, 1996; Pintore et al., 2006). This problem can be solved explicitly and can be shown to approximate early stopping. To be more specific, the solution to following optimization problem approximates the output function of the network after gradient descent training for $t$ steps with learning rate $\bar{\eta}/n$:

$$\min_{g \in C^2(S)} \sum_{j=1}^M [g(x_j) - y_j]^2 + \frac{1}{\bar{\eta}t} \int_S \frac{1}{\zeta(x)}(g''(x) - f''(x, \theta_0))^2 \, dx. \quad (7)$$

**Related works** Zhang et al. (2019) described the implicit bias of gradient descent in the kernel regime as minimizing a kernel norm from initialization, subject to fitting the training data. Our result can be regarded as making the kernel norm explicit, thus providing an interpretable description of the bias in function space and further illuminating the role of the parameter initialization procedure. We prove the equivalence in Appendix M. Savarese et al. (2019) showed that infinite-width networks with 2-norm weight regularization represent functions with smallest 1-norm of the second derivative, an example of which are linear splines. We discuss this in Appendix C.4. A recent preprint further develops this direction for two-layer networks with certain activation functions that interpolate data while minimizing a weight norm (Parhi and Nowak, 2019). In contrast, our result characterizes the solutions of training from a given initialization without explicit regularization, which turn out to minimize a weighted 2-norm of the second derivative and hence correspond to cubic splines. In finishing this work we became aware of a recent preprint (Heiss et al., 2019) which discusses ridge weight penalty, adaptive splines, and early stopping for one-input ReLU networks training only the output layer. Williams et al. (2019) showed a similar result in the kernel regime for shallow ReLU networks where they train only the second layer and from zero initialization. In contrast, we consider the initialization of the second layer and show that the difference from the initial output function is implicitly regularized by gradient descent. We show the result of training both layers and prove that it can be approximated by training only the second layer in Theorem 4. In addition, we give the explicit form of $\zeta$ in Theorem 9, while the $\zeta$ given by Williams et al. (2019) has a minor error because of a typo in their computation. Most importantly, our statement can be generalized to multivariate regression, different activation functions, training trajectories.

## 4 WIDE NETWORKS AND PARAMETER SPACE

### 4.1 IMPLICIT BIAS IN PARAMETER SPACE FOR A LINEARIZED MODEL

In this section we describe how training a linearized network or a wide network by gradient descent leads to solutions that are biased, having parameter values close to the values at initialization. First, we consider the following linearized model:

$$f^{\text{lin}}(x, \omega) = f(x, \theta_0) + \nabla_\theta f(x, \theta_0)(\omega - \theta_0). \quad (8)$$

We write $\omega$ for the parameter of the linearized model, in order to distinguish it from the parameter of the nonlinearized model. The empirical loss of the linearized model is defined by $L^{\text{lin}}(\omega) = \sum_{j=1}^M \ell(f^{\text{lin}}(x_j, \omega), y_j)$. The gradient descent iteration for the linearized model is given by

$$\omega_0 = \theta_0, \quad \omega_{t+1} = \omega_t - \eta\nabla_\theta f(\mathcal{X}, \theta_0)^T \nabla_{f^{\text{lin}}(\mathcal{X}, \omega_t)} L^{\text{lin}}. \quad (9)$$

Next, we consider wide neural networks. According to Lee et al. (2019, Theorem H.1),

$$\sup_t \|f^{\mathrm{lin}}(x,\omega_t) - f(x,\theta_t)\|_2 = O(n^{-\frac{1}{2}})$$

with arbitrarily high probability. So gradient descent training of a wide network or of the linearized model give similar trajectories and solutions in function space. Both fit the training data perfectly, meaning $f^{\mathrm{lin}}(\mathcal{X},\omega_\infty) = f(\mathcal{X},\theta_\infty) = \mathcal{Y}$, and are also approximately equal outside the training data.

Under the assumption that $\mathrm{rank}(\nabla_\theta f(\mathcal{X},\theta_0)) = M$, the gradient descent iterations (9) converge to the unique global minimum that is closest to initialization (Gunasekar et al., 2018a; Zhang et al., 2019), which is the solution of following constrained optimization problem (further details and remarks are provided in Appendix E):

$$\min_\omega \|\omega - \theta_0\|_2 \quad \text{s.t. } f^{\mathrm{lin}}(\mathcal{X},\omega) = \mathcal{Y}. \tag{10}$$

## 4.2 TRAINING ONLY THE OUTPUT LAYER APPROXIMATES TRAINING ALL PARAMETERS

From now on we consider networks with a single hidden layer of $n$ ReLUs and a linear output $f(x,\theta) = \sum_{i=1}^n W_i^{(2)}[W_i^{(1)}x + b_i^{(1)}]_+ + b^{(2)}$. We show that the functions and parameter vectors obtained by training the linearized model are close to those obtained by training only the output layer. Hence, by the arguments of the previous section, training all parameters of a wide network or training only the output layer gives similar functions.

Let $\theta_0 = \mathrm{vec}(\overline{W}^{(1)}, \overline{b}^{(1)}, \overline{W}^{(2)}, \overline{b}^{(2)})$ be the parameter at initialization so that $f^{\mathrm{lin}}(\cdot,\theta_0) = f(\cdot,\theta_0)$. After training the linearized network let the parameter be $\omega_\infty = \mathrm{vec}(\widehat{W}^{(1)}, \widehat{b}^{(1)}, \widehat{W}^{(2)}, \widehat{b}^{(2)})$. Using initialization (2), with probability arbitrarily close to 1, $\overline{W}_i^{(1)}, \overline{b}_i^{(1)} = O(1)$ and $\overline{W}_i^{(2)}, \overline{b}^{(2)} = O(n^{-\frac{1}{2}})$.[1] Therefore, writing $H$ for the Heaviside function, we have

$$\nabla_{W_i^{(1)},b_i^{(1)}} f(x,\theta_0) = \left[\overline{W}_i^{(2)} H(\overline{W}_i^{(1)}x + \overline{b}^{(1)}) \cdot x, \overline{W}_i^{(2)} H(\overline{W}_i^{(1)}x + \overline{b}_i^{(1)})\right] = O(n^{-\frac{1}{2}}),$$

$$\nabla_{W_i^{(2)},b^{(2)}} f(x,\theta_0) = \left[[\overline{W}_i^{(1)}x + \overline{b}_i^{(1)}]_+, 1\right] = O(1). \tag{11}$$

So when $n$ is large, if we use gradient descent with a constant learning rate for all parameters, then the changes of $W^{(1)}, b^{(1)}, b^{(2)}$ are negligible compared with the changes of $W^{(2)}$. So approximately we can train just the output weights, $W_i^{(2)}, i=1,...,n$, and fix all other parameters. This corresponds to a smaller linear model. Let $\widetilde{\omega}_t = \mathrm{vec}(\overline{W}_t^{(1)}, \overline{b}_t^{(1)}, \widetilde{W}_t^{(2)}, \overline{b}_t^{(2)})$ be the parameter at time $t$ under the update rule where $\overline{W}^{(1)}, \overline{b}^{(1)}, \overline{b}^{(2)}$ are kept fixed at their initial values, and

$$\widetilde{W}_0^{(2)} = \overline{W}^{(2)}, \quad \widetilde{W}_{t+1}^{(2)} = \widetilde{W}_t^{(2)} - \eta \nabla_{W^{(2)}} L^{\mathrm{lin}}(\widetilde{\omega}_t). \tag{12}$$

Let $\widetilde{\omega}_\infty = \lim_{t\to\infty} \widetilde{\omega}_t$. By the above discussion, we expect that $f^{\mathrm{lin}}(x,\widetilde{\omega}_\infty)$ is close to $f^{\mathrm{lin}}(x,\omega_\infty)$. In fact, we prove the following for the MSE loss. The proof and further remarks are provided in Appendix F. We relate Theorem 4 to training a wide network in Appendix G.

**Theorem 4** (Training only output weights vs linearized network). *Consider a finite data set $\{(x_i,y_i)\}_{i=1}^M$. Assume that (1) we use the MSE loss $\ell(\widehat{y},y) = \frac{1}{2}\|\widehat{y} - y\|_2^2$; (2) $\inf_n \lambda_{\min}(\hat{\Theta}_n) > 0$. Let $\omega_t$ denote the parameters of the linearized model at time $t$ when we train all parameters using (9), and let $\widetilde{\omega}_t$ denote the parameters at time $t$ when we only train weights of the output layer using (12). If we use the same learning rate $\eta$ in these two training processes and $\eta < \frac{2}{n\lambda_{\max}(\hat{\Theta}_n)}$, then for any $x \in \mathbb{R}$, with probability arbitrarily close to 1 over the random initialization (2),*

$$\sup_t |f^{\mathrm{lin}}(x,\widetilde{\omega}_t) - f^{\mathrm{lin}}(x,\omega_t)| = O(n^{-1}), \text{ as } n \to \infty. \tag{13}$$

*Moreover, in terms of the parameter trajectories we have $\sup_t \|\overline{W}_t^{(1)} - \widehat{W}_t^{(1)}\|_2 = O(n^{-1})$, $\sup_t \|\overline{b}_t^{(1)} - \widehat{b}_t^{(1)}\|_2 = O(n^{-1})$, $\sup_t \|\widetilde{W}_t^{(2)} - \widehat{W}_t^{(2)}\|_2 = O(n^{-3/2})$, $\sup_t \|\overline{b}_t^{(2)} - \widehat{b}_t^{(2)}\| = O(n^{-1})$.*

In view of the arguments in this section, in the next sections we will focus on training only the output weights and understanding the corresponding solution functions.

---

[1] More precisely, for any $\delta > 0$, $\exists C$, s.t. with prob. $1 - \delta$, $|\overline{W}_i^{(2)}|, |\overline{b}^{(2)}| \leq Cn^{-1/2}$ and $|\overline{W}_i^{(1)}|, |\overline{b}_i^{(1)}| \leq C$.

## 5 GRADIENT DESCENT LEADS TO SIMPLE FUNCTIONS

In this section we provide a function space characterization of the implicit bias previously described in parameter space. According to (10), gradient descent training of the output weights (12) achieves zero loss, $f^{\text{lin}}(x_j, \widetilde{\omega}_\infty) - f^{\text{lin}}(x_j, \theta_0) = \sum_{i=1}^n (\widetilde{W}_i^{(2)} - \overline{W}_i^{(2)})[\overline{W}_i^{(1)} x_j + \overline{b}_i]_+ = y_j - f(x_j, \theta_0), j = 1, ..., M$, with minimum $\|\widetilde{W}^{(2)} - \overline{W}^{(2)}\|_2^2$. Hence gradient descent is actually solving

$$\min_{W^{(2)}} \|W^{(2)} - \overline{W}^{(2)}\|_2^2 \quad \text{s.t.} \quad \sum_{i=1}^n (W_i^{(2)} - \overline{W}_i^{(2)})[W_i^{(1)} x_j + b_i]_+ = y_j - f(x_j, \theta_0), j = 1, ..., M. \quad (14)$$

To simplify the presentation, in the following we let $f^{\text{lin}}(x, \theta_0) \equiv 0$ by using the ASI trick (see Appendix C.2). The analysis still goes through without this.

### 5.1 INFINITE WIDTH LIMIT

We reformulate problem (14) in a way that allows us to consider the limit of infinitely wide networks, with $n \to \infty$, and obtain a deterministic counterpart, analogous to the convergence of the NTK. Let $\mu_n$ denote the empirical distribution of the samples $(W_i^{(1)}, b_i)_{i=1}^n$, so that $\mu_n(A) = \frac{1}{n} \sum_{i=1}^n \mathbb{1}_A\left((W_i^{(1)}, b_i)\right)$. Here $\mathbb{1}_A$ is the indicator function for measurable subsets A in $\mathbb{R}^2$. We further consider a function $\alpha_n : \mathbb{R}^2 \to \mathbb{R}$ whose value encodes the difference of the output weight from its initialization for a hidden unit with input weight and bias given by the argument, $\alpha_n(W_i^{(1)}, b_i) = n(W_i^{(2)} - \overline{W}_i^{(2)})$. Then (14) with ASI can be rewritten as

$$\min_{\alpha_n \in C(\mathbb{R}^2)} \int_{\mathbb{R}^2} \alpha_n^2(W^{(1)}, b) \, \mathrm{d}\mu_n(W^{(1)}, b) \text{ s.t.} \int_{\mathbb{R}^2} \alpha_n(W^{(1)}, b)[W^{(1)} x_j + b]_+ \, \mathrm{d}\mu_n(W^{(1)}, b) = y_j, \quad (15)$$

where $j$ ranges from 1 to $M$. Here we minimize over functions $\alpha_n$ in $C(\mathbb{R}^2)$, but since only the values on $(W_i^{(1)}, b_i)_{i=1}^n$ are taken into account, we can take any continuous interpolation of $\alpha_n(W_i^{(1)}, b_i)$, $i = 1, ..., n$. Now we can consider the infinite width limit. Let $\mu$ be the probability measure of $(\mathcal{W}, \mathcal{B})$. We obtain a continuous version of problem (15) by substituting $\mu$ for $\mu_n$. Since we know that $\mu_n$ weakly converges to $\mu$, we prove that in fact the solution of problem (15) converges to the solution of the continuous problem, which is formulated in the following theorem. Details in Appendix H.

**Theorem 5.** *Let $(W_i^{(1)}, b_i)_{i=1}^n$ be i.i.d. samples from a pair $(\mathcal{W}, \mathcal{B})$ of random variables with finite fourth moment. Suppose $\mu_n$ is the empirical distribution of $(W_i^{(1)}, b_i)_{i=1}^n$ and $\overline{\alpha}_n(W^{(1)}, b)$ is the solution of (15). Let $\overline{\alpha}(W^{(1)}, b)$ be the solution of the continuous problem with $\mu$ in place of $\mu_n$. Then for any bounded $[-L, L]$, $\sup_{x \in [-L,L]} |g_n(x, \overline{\alpha}_n) - g(x, \overline{\alpha})| = O(n^{-1/2})$ with high probability, where $g_n(x, \alpha_n) = \int_{\mathbb{R}^2} \alpha_n(W^{(1)}, b)[W^{(1)} x + b]_+ \, \mathrm{d}\mu_n(W^{(1)}, b)$ is the function represented by a network with $n$ hidden neurons after training, and $g(x, \alpha) = \int_{\mathbb{R}^2} \alpha(W^{(1)}, b)[W^{(1)} x + b]_+ \, \mathrm{d}\mu(W^{(1)}, b)$ is the function represented by the infinite-width network.*

### 5.2 FUNCTION SPACE DESCRIPTION OF THE IMPLICIT BIAS

Next we connect the problem from the previous section to second derivatives by first rewriting it in terms of breakpoints. Consider the breakpoint $c = -b/W^{(1)}$ of a ReLU with weight $W^{(1)}$ and bias $b$. We define a corresponding random variable $\mathcal{C} = -\mathcal{B}/\mathcal{W}$ and let $\nu$ denote the distribution of $(\mathcal{W}, \mathcal{C})$.[2] Then with $\gamma(W^{(1)}, c) = \alpha(W^{(1)}, -cW^{(1)})$ the continuous version of (15) is equivalently given as

$$\min_{\gamma \in C(\mathbb{R}^2)} \int_{\mathbb{R}^2} \gamma^2(W^{(1)}, c) \, \mathrm{d}\nu(W^{(1)}, c) \text{ s.t.} \int_{\mathbb{R}^2} \gamma(W^{(1)}, c)[W^{(1)}(x_j - c)]_+ \, \mathrm{d}\nu(W^{(1)}, c) = y_j, \quad (16)$$

where $j$ ranges from 1 to $M$. Let $\nu_{\mathcal{C}}$ denote the distribution of $\mathcal{C} = -\mathcal{B}/\mathcal{W}$, and $\nu_{\mathcal{W}|\mathcal{C}=c}$ the conditional distribution of $\mathcal{W}$ given $\mathcal{C} = c$. Suppose $\nu_{\mathcal{C}}$ has support $\text{supp}(\nu_{\mathcal{C}})$ and a density function $p_{\mathcal{C}}(c)$.

---

[2] Here we assume that $\mathbb{P}(\mathcal{W} = 0) = 0$ so that the random variable $\mathcal{C}$ is well defined. It is not an important restriction, since neurons with weight $W^{(1)} = 0$ give constant functions that can be absorbed in the bias of output layer.

Let $g(x,\gamma) = \int_{\mathbb{R}^2} \gamma(W^{(1)},c)[W^{(1)}(x-c)]_+ \, \mathrm{d}\nu(W^{(1)},c)$, which again corresponds to the output function of the network. Then, the second derivative $g''$ with respect to $x$ (see Appendix I) satisfies $g''(x,\gamma) = p_{\mathcal{C}}(x) \int_{\mathbb{R}} \gamma(W^{(1)},x) |W^{(1)}| \, \mathrm{d}\nu_{\mathcal{W}|\mathcal{C}=x}(W^{(1)})$. Thus $\gamma(W^{(1)},c)$ is closely related to $g''(x,\gamma)$ and we can try to express (16) in terms of $g''(x,\gamma)$. Since $g''(x,\gamma)$ determines $g(x,\gamma)$ only up to linear functions, we consider the following problem:

$$\min_{\gamma \in C(\mathbb{R}^2), u \in \mathbb{R}, v \in \mathbb{R}} \quad \int_{\mathbb{R}^2} \gamma^2(W^{(1)},c) \, \mathrm{d}\nu(W^{(1)},c)$$

$$\text{subject to} \quad ux_j + v + \int_{\mathbb{R}^2} \gamma(W^{(1)},c)[W^{(1)}(x_j-c)]_+ \, \mathrm{d}\nu(W^{(1)},c) = y_j, \quad j = 1,...,M. \tag{17}$$

Here $u,v$ are not included in the cost. They add a linear function to the output of the neural network. If $u$ and $v$ in the solution of (17) are small, then the solution is close to the solution of (16). Ongie et al. (2020) also use this trick to simplify the characterization of neural networks in function space. Next we study the solution of (17) in function space. This is our main technical result.

**Theorem 6** (Implicit bias in function space). *Assume $\mathcal{W}$ and $\mathcal{B}$ are random variables with $\mathbb{P}(\mathcal{W} = 0) = 0$, and let $\mathcal{C} = -\mathcal{B}/\mathcal{W}$. Let $\nu$ denote the probability distribution of $(\mathcal{W},\mathcal{C})$. Suppose $(\overline{\gamma},\overline{u},\overline{v})$ is the solution of* (17), *and consider the corresponding output function*

$$g(x,(\overline{\gamma},\overline{u},\overline{v})) = \overline{u}x + \overline{v} + \int_{\mathbb{R}^2} \overline{\gamma}(W^{(1)},c)[W^{(1)}(x-c)]_+ \, \mathrm{d}\nu(W^{(1)},c). \tag{18}$$

*Let $\nu_{\mathcal{C}}$ denote the marginal distribution of $\mathcal{C}$ and assume it has a density function $p_{\mathcal{C}}$. Let $\mathbb{E}(\mathcal{W}^2|\mathcal{C})$ denote the conditional expectation of $\mathcal{W}^2$ given $\mathcal{C}$. Consider the function $\zeta(x) = p_{\mathcal{C}}(x)\mathbb{E}(\mathcal{W}^2|\mathcal{C}=x)$. Assume that training data $x_i \in \mathrm{supp}(\zeta)$, $i=1,...,m$. Consider the set $S = \mathrm{supp}(\zeta) \cap [\min_i x_i, \max_i x_i]$. Then $g(x,(\overline{\gamma},\overline{u},\overline{v}))$ satisfies $g''(x,(\overline{\gamma},\overline{u},\overline{v})) = 0$ for $x \notin S$ and for $x \in S$ it is the solution of the following problem:*

$$\min_{h \in C^2(S)} \int_S \frac{(h''(x))^2}{\zeta(x)} \, \mathrm{d}x \quad s.t. \quad h(x_j) = y_j, \quad j = 1,...,m. \tag{19}$$

The proof is provided in Appendix I, where we also present the corresponding statement without ASI. We study the explicit form of this function in the next section.

### 5.3 Explicit form of the curvature penalty function

**Proposition 7.** *Let $p_{\mathcal{W},\mathcal{B}}$ denote the joint density function of $(\mathcal{W},\mathcal{B})$ and let $\mathcal{C} = -\mathcal{B}/\mathcal{W}$ so that $p_{\mathcal{C}}$ is the breakpoint density. Then $\zeta(x) = \mathbb{E}(W^2|C=x)p_{\mathcal{C}}(x) = \int_{\mathbb{R}} |W|^3 p_{\mathcal{W},\mathcal{B}}(W,-Wx) \, \mathrm{d}W$.*

The proof is presented in Appendix J. If we allow the initial weight and biases to be sampled from a suitable joint distribution, we can make the curvature penalty $\rho = 1/\zeta$ arbitrary.

**Proposition 8** (Constructing any curvature penalty). *Given any function $\varrho : \mathbb{R} \to \mathbb{R}_{>0}$, satisfying $Z = \int_{\mathbb{R}} \frac{1}{\varrho} < \infty$, if we set the density of $\mathcal{C}$ as $p_{\mathcal{C}}(x) = \frac{1}{Z}\frac{1}{\varrho(x)}$ and make $\mathcal{W}$ independent of $\mathcal{C}$ with non-vanishing second moment, then $(\mathbb{E}(W^2|C=x)p_{\mathcal{C}}(x))^{-1} = (\mathbb{E}(W^2)p_{\mathcal{C}}(x))^{-1} \propto \varrho(x)$, $x \in \mathbb{R}$.*

Further remarks on sampling and independent variables are provided in Appendix J. To conclude this section we compute the explicit form of $\zeta$ for several common initialization procedures.

**Theorem 9** (Explicit form of the curvature penalty for common initializations).

(a) *Gaussian initialization. Assume that $\mathcal{W}$ and $\mathcal{B}$ are independent, $\mathcal{W} \sim \mathcal{N}(0,\sigma_w^2)$ and $\mathcal{B} \sim \mathcal{N}(0,\sigma_b^2)$. Then $\zeta$ is given by $\zeta(x) = \frac{2\sigma_w^3\sigma_b^3}{\pi(\sigma_b^2 + x^2\sigma_w^2)^2}$.*

(b) *Binary-uniform initialization. Assume that $\mathcal{W}$ and $\mathcal{B}$ are independent, $\mathcal{W} \in \{-1,1\}$ and $\mathcal{B} \sim \mathcal{U}(-a_b,a_b)$ with $a_b \geq L$. Then $\zeta$ is constant on $[-L,L]$.*

(c) *Uniform initialization. Assume that $\mathcal{W}$ and $\mathcal{B}$ are independent, $\mathcal{W} \sim \mathcal{U}(-a_w, a_w)$ and $\mathcal{B} \sim \mathcal{U}(-a_b,a_b)$ with $\frac{a_b}{a_w} \geq L$. Then $\zeta$ is constant on $[-L,L]$.*

The proof is provided in Appendix K. Theorem 9 (b) and (c) show that for certain distributions of $(\mathcal{W},\mathcal{B})$, $\zeta$ is constant. In this case problem (19) is solved by the cubic spline interpolation of the data with natural boundary conditions (Ahlberg et al., 1967). The case of general $\zeta$ is solved by space adaptive natural cubic splines, which can be computed numerically by solving a linear system and theoretically in an RKHS formalism. We provide details in Appendix O.

## 6 CONCLUSION AND DISCUSSION

We obtained a explicit description of the implicit bias of gradient descent for mean squared error regression with wide shallow ReLU networks. We presented a result for the univariate case and generalizations to multi-variate ReLU networks and networks with different activation functions. Our result can also help us characterize the training trajectory of gradient descent in function space.

Our main result shows that the trained network outputs a function that interpolates the training data and has the minimum possible weighted 2-norm of the second derivative with respect to the input. This corresponds to an spatially adaptive interpolating spline. The space of interpolating splines is a linear space which has a dimension that is linear in the number of data points. Hence our result means that, even if the network has many parameters, the complexity of the trained functions will be adjusted to the number of data points. Interpolating splines have been studied in great detail in the literature and our result allows us to directly apply corresponding generalization results to the case of trained networks. This is related to approximation theory and characterizations for the number of samples and their spacing needed in order to approximate functions from a given smoothness class to a desired precision (Rieger and Zwicknagl, 2010; Wendland, 2004).

Zhang et al. (2019) described the implicit bias of gradient descent as minimizing a RKHS norm from initialization. Our result can be regarded as making the RKHS norm explicit, thus providing an interpretable description of the bias in function space. Compared with Zhang et al. (2019), our results give a precise description of the role of the parameter initialization scheme, which determines the inverse curvature penalty function $\zeta$. This gives us a rather good picture of how the initialization affects the implicit bias of gradient descent. This could be used in order to select a good initialization scheme. For instance, one could conduct a pre-assessment of the data to estimate the locations of the input space where the target function has a high curvature, and choose the parameter initialization accordingly. This is an interesting possibility to experiment with, based on our theoretical result.

Our result can also be interpreted in combination with early stopping. The training trajectory is approximated by a smoothing spline, meaning that the network will filter out high frequencies which are usually associated to noise in the training data. This behaviour is sometimes referred to as a spectral bias (Rahaman et al., 2019).

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
