# OpenReview forum: "Implicit bias of gradient descent for mean squared error regression with wide neural networks"
_ICLR.cc/2021/Conference — Reject_

### Official Review · AnonReviewer3 · 2020-10-14
**Somewhat incremental contribution? And the paper organization can be clearly improved.**

**Rating:** 5
**Confidence:** 3

**Review:**

**Summary**:
In this article, the authors characterized the implicit bias of gradient descent-based learning in the setting of wide single-hidden-layer neural networks with ReLU activation. More precisely, it was shown in Theorem 1 that the trained network output is "close" to that of the zero-training-error solution that is the "closest" to initialization when the number of neurons $n$ is large. In particular, such a "closest" solution is defined via the so-called *curvature penalty function* $1/\zeta$ that depends on the random initialization of the parameters. Theses results were shown in Theorem 1 to hold for *scalar* input with ReLU activation, and then generalized to *multi-dimensional* input with ReLU nonlinearity in Theorem 2 and to general homogeneous activation function in Corollary 3.

**Strong points**:
This paper provides a precise characterization of the implicit bias of (full-batch) gradient descent method in training single-hidden-layer NN, by studying the resulting solution in the setting of large network width. The major contribution of this work, I believe, is to provide *explicit* characterization of the impact of the distribution of the random initialization on the resulting solution, which is then connected to cubic spline interpolation.

**Weak points**:
My first concern is that the proposed analysis seems somewhat incremental (compared to previous efforts discussed in P4) and fails to provide sufficiently novel insights on the implicit bias of gradient descent: it is good to have the explicit form as in (1) that depends on the random initialization and the second derivative, and I believe it worth more than a single paragraph of discussion and a single figure to illustrate its practical implications, e.g., how does the number of training sample $M$ come into play? Can similar behaviors be empirically observed beyond the single-hidden-layer model, even without proof? how should we choose the initialization scheme in different problems?

My second concern is the organization of the paper: I do not understand why the authors have chosen to present the long and complex theorems in Section 2 before introducing the models and notations in Section 3, this makes the paper, at least for me, much harder to read and follow. Also, I do not understand why the authors have chosen to present the model and notations in Sec 3.1 and 3.2 in the general context of $L$-hidden-layer neural networks: none of the technical results are concerned with this "deep" case and I personally find this only creates unnecessary confusion.

**Recommendation**:
I find this paper borderline, and according to the weak points mentioned above, I'm more leaning toward a reject.

**Detailed comments**:
* abstract: "of the second derivative": the second derivative of what with respect to what?
* Theorem 1: "for which $f(x, \theta^*)$ attains zero training error": $f(\cdot)$ is not yet defined, and it would also be helpful to recall the definition of $C^2(S)$ in (1).
* Below (9): it would be helpful to clarify the conditions under which Lee et al. (2019, Theorem H.1) hold and if they are compatible with the assumptions for instance in Theorem 1 of the present article.
* Above (10): "gradient descent training of a wide network or of the linearized model giveS similar trajectories and solutions in function space": the argument on the "trajectory" is not reflected in the last equation in P5, which only characterizes the network output.
* Above (10): "converge to the unique global minimum": how is the uniqueness ensured here?
* Above (11) footnote 1: is this a claim? If yes, it would be helpful to point out in which Section of the appendix is this proved.
* After (11): not sure to understand why the change in $b^{(2)}$ is also negligible compared to that of $W^{(2)}$.
* Theorem 4: it would be helpful to clarify whether $\lambda_\max(\hat \Theta_n)$ is of order $O(1)$ with respect to $n$, that is, does $\eta < \frac{2}{n \lambda_\max(\hat \Theta_n)}$ mean the the step size should scale like $O(n^{-1})$ in the $n \to \infty$ limit?

---

> ### Author Response · Authors · 2020-11-14
> **Thank you for your valuable review and comments.**
>
> Thank you for your valuable review and comments. Here let us respond to your concerns:
>
> 1. You mentioned a concern that the result might be somewhat incremental. The implicit bias of gradient descent is one of central topics in deep learning theory and there are many recent articles on the subject. Our result offers a clear description of the role of the initialization distribution and an explicit expression for the bias in function space. Moreover, we provide the first result for multidimensional inputs as well as extensions to training trajectories and other activation functions. We think these advances are quite significant, considering that each of these extensions could warrant an individual publication. The proof technique that we present is extremely versatile, as illustrated by the several results that we have been able to obtain from it, and, in contrast to other previous approaches to the implicit bias, we think it can also be applied to even more general settings beyond the kernel regime, a topic we are currently investigating.
>
> 2.    You ask about the implications of our main result. Our result shows that the output function of the network is captured by interpolating splines. The space of interpolating splines is a linear space which has a dimension that is linear in the number of data points. Hence our result means that, even if the network has many more parameters than the number of data points it is trained with, the complexity of the trained functions will be adjusted to the number of data points. This function class has been studied in great detail in the literature and our result allows us to directly apply generalization results for spline interpolation to the case of trained networks. This is related to approximation theory, where the number of samples and their spacing, which are needed in order to approximate a function from a given smoothness class, are characterized. Two useful references about this are [1],[2].
>
>    I can try to give you some more intuition. An easy example are functions that minimize the 1-norm of the second derivative. In this case, we end up getting the linear interpolation of the training samples. Suppose the fill distance of the training set is h (defined by $\max_{x\in\Omega}\min_{x_j\in X} \|x-x_j\|$, $\Omega$ is domain of the function and $X$ is the training set), then the interpolating function approximates the ground truth with an error of O(h) under the assumption that the ground truth is a function with bounded first derivative. Increasing the number of training samples decreases the fill distance h, so that we have better generalization.
>
>    Regarding the role of the curvature penalty function in generalization, our opinion is that this will depend on the properties of the ground truth. We can use a higher curvature penalty in order to obtain a smoother function at a given location. As we show, this can be achieved by tuning the initialization distribution. For standard initialization, the inverse curvature penalty zeta is 1/(1+x^2)^2, which is peaked at the origin. Hence our result implies that standard initialization biases the solutions of gradient descent to functions which are more linear far from the origin. Our result can also be interpreted in combination with early stopping. Since the training trajectory is approximated by a smoothing spline, in this case the network will filter out high frequencies which are usually associated to noise in the training data.
>
>    In terms of choosing the initialization: If we know that the underlying function has high curvature near the origin and is flat elsewhere, the settings in Figure A2, where zeta is peaked at the origin, is better than a uniform zeta. In principle, one can also imagine to conduct a pre-assessment of the data to estimate the locations of the input space where the target function has a high curvature, and choose the parameter initialization accordingly. This is an interesting possibility to experiment with, based on our theoretical result.
>
> 3. Beyond single layer. About deeper networks, we are not sure about the results for now. Some experiments show that the outputs of deep networks after training are close to the linear interpolation, which means that 1-norm of the second derivative of the output function is minimized. The general sense in the community is that this is related to the adaptive regime (in contrast to the lazy or kernel regime). The adaptive regime is not very well understood at this moment and is the subject of intensive research in numerous groups. We hope to develop methods for that setting in the future.

---

> > ### Author Response · Authors · 2020-11-14
> > **More response**
> >
> > 4. Your second concern is about the organization of the paper. Concretely you mention that the detailed definitions and settings could be presented before the main results. Our rationale for the structure of the paper was that the definitions are standard and readers who are familiar would prefer to read the main results and discussion before technical details. We highlighted the definitions with a subsection title and hyperlinked them from the main result so that they are easily accessible. However, we understand your comment, and we can move some of the settings and definitions to the front, which of course is very easily done.
> >
> > 5. Responses to concrete comments.
> > * We will add "the second derivative of the output function with respect to the input".
> > * Thank you for pointing out this. Actually f is defined inline in Section 3.1, and it is not numbered. We will put the definition of f in display mode and put a hyperlink in the Theorem 1.
> > * It is a good suggestion. The assumptions of Lee et al. (2019, Theorem H.1) can be easily verified. We can include them if we have enough space.
> > * The equation you mentioned actually characterizes the training  trajectory because $\sup_t$ take the supreme over all training time. So it means that at any time, the output of network can be approximated by a linear model.
> > * The uniqueness of global minimum is ensured because the objective of the optimization problem is strictly convex and the constraints are linear.
> > * The result in footnote 1 can be gotten from the initialization scheme straightforwardly. The reason is because the network has 1 input and n hidden units. So weights in second layer is scaled by $\sqrt{1/n}$ and weights in first layer is scaled by 1.
> > * The reason is $b^{(2)}$ only has one parameter while $W^{(2)}$ has n parameters. When n goes to infinity, the contribution of $b^{(2)}$ is negligible.
> > * $\lambda_{\max}(\hat{\Theta}_n)$ is of order $O(1)$ because the empirical NTK converges in probability. We discuss it in Appendix C.1. And you statement is correct. The step size does scale like $O(n^{-1})$.
> >
> > [1] C. Rieger and B. Zwicknagl. Sampling inequalities for infinitely smooth functions, with applications to interpolation and machine learning. Advances in Computational Mathematics, 32(1):103, 2010.
> >
> > [2] H. Wendland. Scattered data approximation, 2004.

---

### Official Review · AnonReviewer2 · 2020-10-28
**Too packed with Maths and abrupt**

**Rating:** 6
**Confidence:** 1

**Review:**

Update after the rebuttal: I would like to thank the authors for the detailed reply and for addressing raised issues in the submission. I appreciate the authors' rationale, but the "standard" structure of papers makes it is easier to follow. The same for a conclusion, for the authors it may be reiterating the same ideas, but personally I found conclusions the best place where one can quickly get a flavour what has been done in the paper to assess whether it is actually worth spending time reading it in details. Also, they are helpful in cases like this when a reader (me) is outside of the research field of the paper. Regarding discussions, I appreciate that there is discussion for Theorem 1, but there are theorems and propositions stated in the formal language only which would benefit by being repeated in plain English. If they are just technical results required for the main proof, they may be moved to appendix then.
Overall, I am increasing my score to reflect positive changes in the submission.

There is a minor mistakes:
- In the first line of Conclusion: "obtained aN explicit"


=========================================================================================================


The paper considers the implicit bias (i.e. why neural networks generalise well) in gradient descent learning of wide neural networks for the regression problem. The theoretical results are first stated for wide ReLU network for a 1D regression problem. They are then generalised for multivariate regression and different activation functions.

It is difficult for me to assess the main content of the paper, as it is outside of my comfort zone. Therefore, this review would be mostly a feedback on overall structure and clarity of the paper.

Strong points:
* apparent solid and large theoretical analysis
* addressing the important problem of understanding why the neural networks and gradient descent lead to such successful results in various domains

Weak points:
* the paper is not friendly for outsiders of the particular research avenue. It is packed with content without too much space to discussion of presented results to the point that the conclusion section is missing in the paper
* the structure of the paper is very odd which leads to future references that appear only 3 pages afterwards (see details below)

As mentioned I can't assess the main content of the paper, but with my educational guess I would recommend to reject the paper in the current version. Careful revision is required to make it a complete piece of work (add conclusion and discussion) and make it more accessible for wider machine learning audience.

In particular, for the above mentioned weak points:
* I appreciate space constrains, but probably some of the material can be moved to supplementary completely to allow discussions of the main results more. The theoretical findings are mostly presented in the formal language and there is a lack of plain English discussion on what this means and how it affect the bigger picture. And the paper has to have conclusion section and shouldn't end abruptedly
* It seems that Section 2 and 3 should be swapped as I cannot see the reason why notations and problem formulation are presented after the main results: for those who are inside the field and can understand Section 2 without introduction would not need then introduction in Section 3 at all. And those who do need Section 3 would not understand Section 2 without it.
This also leads to these inconvenient future references. E.g., referring in Theorem (1) to eq.(5) that appears 3 pages after that is a questionable choice.

Some other suggestions/concerns:
1.	ReLU is introduced in the last paragraph in Introduction, but used in the previous paragraph
2.	NTK is defined well after it is used for the first time
3.	The last sentence in Section 2 – there was nothing before about different training trajectories
4.	Notation clash: in section 2 sigma denotes an activation function (different from ReLU) and in section 3 sigma denotes a parameter of initialisation distribution: Gaussian and uniform
5.	After eq. (7): “We will use subscript i to index neurons and subscript t to index time”, i is also used for training points.
6.	Strictly speaking n in equation between eq. (9) and (10) is not well-defined
7.	ASI is not defined

---

> ### Author Response · Authors · 2020-11-14
> **Thank you for taking the time to review our manuscript**
>
> Thank you for taking the time to review our manuscript, especially considering it as being outside of your comfort zone, and commenting on the structure and clarity. In the following we respond to your comments:
>
> You mentioned that the paper might not be friendly for outsiders of the particular research avenue.
>
> Please consider that the structure and language of the manuscript is reflective of the fast pace and increasing complexity of the topics that we are dealing with in the community. The implicit bias of gradient descent in training neural networks has become a central topic of research in theoretical deep learning which has garnered numerous recent publications. With this in mind, our motivation for the structure of the paper was that readers would prefer to read a crisp presentation of the main results and discussion first, before going into technical details or revisiting frequent definitions. We highlighted the definitions with a subsection title and hyperlinked them from the main results so that they are visible and easily accessible. Contrary to one of your comments, we think a minimum of definitions is still needed even for readers who are familiar with the literature, just that the details can be presented after. Of course we can move some of the definitions to the front, but we wanted to explain our rationale nonetheless.
>
> Kindly note that we actually do have a discussion section, namely section 2 Main Results and Discussion. As you observed, in instead of repeating the main points in a conclusion section, we used the available space to communicate supporting results and key derivations. This is actually not so uncommon. However, we are aware that some readers might find a reiteration of the main points useful, and with the additional space we can gladly add a short conclusion section.
>
> With the additional space we will gladly move some of the definitions to the front and add plain text explanations and a short conclusion section, which of course is very easily done.
>
>
> In regard to your specific comments:
> 1. Definition of ReLU. Ok, we can add the definition at the first use.
> 2. Definition of the NTK initialization and NTK parametrization. Ok, we will add a pointer to this.
> 3. Training trajectories. Notice that these are actually discussed in eq. (4) and the preceding paragraph.
> 4. Notation clash sigma. Good point, sigma is standard notation for the standard deviation of a random variable, and it is also standard notation for an activation function. We will remove the clash.
> 5. Index i, you are correct, thanks for pointing this out. We used j to index training samples but there are a few places where we forget to use j instead of i.
> 6. Although n is not explicitly defined, we said in the beginning of Section 3.3 that the network has n hidden neurons. So n refers to the number of hidden neurons throughout the paper.
> 7. ASI. For this we have a pointer to details in the appendix; see after eq. (14).

---

### Official Review · AnonReviewer1 · 2020-10-28
**Intuitive but interesting insights into initialization and implicit regularization**

**Rating:** 7
**Confidence:** 3

**Review:**

This paper presents a function space view of 2-layer ReLU neural networks and the implicit regularization associated with full-batch gradient descent for various initializations of the weights. In past work, it was shown that for very wide neural networks, the *global* minimizer of a loss plus weight decay regularizer corresponds amounts to regularizing the total variation of the second derivative of a function (and related results in higher dimensions). This paper explores the effective regularization associated with performing gradient descent on an *unregularized* squared error loss. The resulting regularizer is akin to a weighted 2-norm of the second derivative, where the weighting function depends on the distribution of the initial weights. This result addresses an important open problem, provides interesting insights into the role of initialization and implicit bias associated with training, and is supported by nice illustrations.

The literature review is strong and covers much of the relevant literature.

Much of the analysis seems to depend on the optimization occurring in the "kernel regime". It is unclear when this is or is not a reasonable model. This issue is particularly salient in light of the comparison of the results with the work of Savarese et al. Savarese considers explicit 2-norm weight decay regularization. Although the paper under review considers unregularized losses, there are multiple studies showing that gradient descent initialized near zero induces 2-norm regularization. So a natural thought is that the Savarese result would also extend (with some non-trivial technical work) to unregularized settings with gradient descent. With this in mind, I expected to see the Savarese norm as a special case of the results of the paper under review, but this is not the case. In particular, the function space regularization calculated in Savarese is NOT an RKHS norm, while the paper under review claims the function space regularization they find IS a kernel norm. I would like to see a more detailed discussion of this potential discrepancy. Section C.4 does not make this clear.

---

> ### Author Response · Authors · 2020-11-24
> **Thank you for your detailed review and the positive evaluation of our paper**
>
> Thank you for your detailed review and the positive evaluation of our paper. You raised an interesting question regarding the difference between RKHS norm and non-RKHS norm. Here let me explain why our characterization gives the RKHS norm while Savarese gave the non-RKHS-norm, and we will add some discussion in Appendix C.4. The reason why training without weight decay gives RKHS norm is because the training trajectory can be approximated by that of a linear model, which corresponds to a certain RKHS. And for training with weight decay, the weight in the first layer is regularized, so it changes the feature space and we can no longer regard that as a linear model. In some literatures, people empirically show that minimizing non-RKHS norm is better than RKHS norm because of the limitation of linear model and kernel regime. However, as far as I know, there is no theory which shows that non-RKHS-norm could result in better generalization than RKHS norm.

---

### Official Review · AnonReviewer6 · 2020-11-06
**Detailed analysis of implicit bias in wide neural networks**

**Rating:** 7
**Confidence:** 4

**Review:**

######################################################################

1.  Paper Summary

This work analyzes the implicit bias of gradient descent for wide, 1 hidden layer networks used for regression and provides a characterization of this bias in function space.  At a high level, as network width increases, gradient descent leads to a solution given by a variational problem penalizing the product of the curvature and the square of the second derivative.  The proof of this proceeds as follows: (1) The solution given by gradient descent on the network can be approximated by the solution by gradient descent on a linear model. (2) Under the standard initialization, the solution for the linear model can be approximated by that of a smaller linear model (corresponding to training the last layer).  (3) The implicit bias for this problem is linked to an alternate optimization problem. (4) The solution to this problem is given by solution to the the variational problem as network width goes to infinity.

######################################################################

2. Strengths

2.1. The results are presented rigorously and the authors consider a number of generalizations involving (1) alternate distributions for weights/biases (2) univariate and multivariate regression (3) alternate nonlinearities. In particular, an extended discussion of generalizations is provided in Appendix O (some of the discussion points in O.3 - O.5 would be nice to include in a conclusion).

2.2. The empirical evidence presented in the main text and Appendix was very useful in providing an intuitive explanation around Theorem 1.  I also particularly liked that the authors provided an interpretation paragraph on page 2, which among other points addressed the reason for involving a linear adjustment to the training data.

2.3. The authors position their results well relative to a large number of related works.  Appendix C clarified a lot of the points regarding related work in the main text especially regarding comparisons between the results of this work and those of Savarese et al. 2019.

######################################################################


3. Minor Limitations

3.1. While I found this to be an interesting and rigorous work, I feel it would be helpful if the authors could provide a bit more intuition around the technical results for the generalizations.  In particular, I found the interpretation and visualizations presented in the work very helpful for understanding the implicit bias described by Theorem 1. However, it would be helpful if the authors could provide a similar intuition for the multi-variate regression setting.

3.2. I found this work a bit difficult to read through due to the several jumps between equation references.  I think one thing that would help improve the readability is adjusting the Appendix such that related equations in the main text are above the related sections in the Appendix.  One example of this is the need to jump back and forth between equations 15, 16, 17 of the main text in Appendix D.

3.3. I feel the authors could present some of the assumptions more clearly in the main text.  As a quick example, I believe the authors assume in equation (9) that grad_(theta) f(X, theta_0) has rank M (as is done in Appendix E), but unless I'm mistaken, this is not clearly stated in the main text, and it would be nice to understand how this rank requirement relates to the width n of the network (or whether this is not an obvious relationship).

3.4. (Very Minor) I think this work could benefit from a discussion of how the implicit bias identified in this work could connect with generalization.  For example, is there any way to understand which curvature penalties would yield solutions that generalize better?

######################################################################

4. Score and Rationale

I would vote for accepting this paper.  I found the result to be insightful in characterizing the inductive bias of 1 hidden layer fully connected networks used for regression.  The author's present a rigorous analysis, which they complement with a number of empirical and theoretical examples.


######################################################################

5. Minor Comments

5.1. (Very minor style recommendation) The current introduction is a nice summary of related work, but I think the paper would be a bit more readable if the main results and discussion were placed in front of these related works and these works were merged with the other related works section.

5.2. (Minor typo) The last sentence of Appendix C.2 appears to be incomplete.

---

> ### Author Response · Authors · 2020-11-24
> **Thank you for your detailed review and the positive evaluation of our paper**
>
> Thank you for your detailed review and the positive evaluation of our paper. Here is our response to the minor limitations you pointed out:
>
> 1. The visualization of the multi-variate regression is still an ongoing work. Computing the solution to multi-variate variational problem is much more complicated than the 1-d case and we plan to do that in future work.
>
> 2. This is a good point. However, the jumps between references are somehow unavoidable since Appendix D is the proof of our main theorem and combine different parts throughout the whole main text. Nevertheless, we will try to improve the readability of our paper.
>
> 3. You are right. Because of the space constraints, we wrote the general description in the main text and put the theorem along with detailed assumptions in the Appendix. We have improved this in the revision.
> About the relation between $\nabla_\theta f(\mathcal{X}, \theta_0)$ and n (here n refers to the number of parameters in $\theta$ since we use linearized model), we can say that $\nabla_\theta f(\mathcal{X}, \theta_0)=M$ almost surely when $n\geq M$. This is because $\nabla_\theta f(\mathcal{X}, \theta_0)$ is a $M\times n$ matrix. The M rows corresponds to M training samples and they are linearly independent in most cases.
>
> 4. In terms of generalization, our result shows that the output function of the network is captured by interpolating splines. The space of interpolating splines is a linear space which has a dimension that is linear in the number of data points. Hence our result means that, even if the network has many more parameters than the number of data points it is trained with, the complexity of the trained functions will be adjusted to the number of data points. This function class has been studied in great detail in the literature and our result allows us to directly apply generalization results for spline interpolation to the case of trained networks. This is related to approximation theory, where the number of samples and their spacing, which are needed in order to approximate a function from a given smoothness class, are characterized. Two useful references about this are [1],[2].
>
>    I can try to give you some more intuition. An easy example are functions that minimize the 1-norm of the second derivative. In this case, we end up getting the linear interpolation of the training samples. Suppose the fill distance of the training set is h (defined by $\max_{x\in\Omega}\min_{x_j\in X} \|x-x_j\|$, $\Omega$ is domain of the function and $X$ is the training set), then the interpolating function approximates the ground truth with an error of O(h) under the assumption that the ground truth is a function with bounded first derivative. Increasing the number of training samples decreases the fill distance h, so that we have better generalization.
>
>    Regarding the role of the curvature penalty function in generalization, our opinion is that this will depend on the properties of the ground truth. We can use a higher curvature penalty in order to obtain a smoother function at a given location. As we show, this can be achieved by tuning the initialization distribution. For standard initialization, the inverse curvature penalty zeta is 1/(1+x^2)^2, which is peaked at the origin. Hence our result implies that standard initialization biases the solutions of gradient descent to functions which are more linear far from the origin. Our result can also be interpreted in combination with early stopping. Since the training trajectory is approximated by a smoothing spline, in this case the network will filter out high frequencies which are usually associated to noise in the training data.
>
>    In terms of choosing the initialization: If we know that the underlying function has high curvature near the origin and is flat elsewhere, the settings in Figure A2, where zeta is peaked at the origin, is better than a uniform zeta. In principle, one can also imagine to conduct a pre-assessment of the data to estimate the locations of the input space where the target function has a high curvature, and choose the parameter initialization accordingly. This is an interesting possibility to experiment with, based on our theoretical result.
>
>
> Responses to your minor comments:
> 1. You mentioned that it's better to merge two parts of the related works. In our opinion, these two parts of the related works have different purposes. The first part serves as an introduction and they are not quite related to our main theorem  and the second part are closely related to our main theorem. That's why we put our main theorem in between these two parts of related works.
> 2. Thank you for pointing out the typo. We have already fixed it.
>
> [1] C. Rieger and B. Zwicknagl. Sampling inequalities for infinitely smooth functions, with applications to interpolation and machine learning. Advances in Computational Mathematics, 32(1):103, 2010.
>
> [2] H. Wendland. Scattered data approximation, 2004.

---

### Official Review · AnonReviewer5 · 2020-11-06
**A characterization of the implicit bias of gradient descent on regression problems and two-layer networks**

**Rating:** 5
**Confidence:** 3

**Review:**

This paper analyzes the implicit bias of gradient descent on a wide two-layer network with standard parameterization and initialization and the squared loss. It is first proved that in this setting, gradient descent on both layers is close to gradient descent on the second layer. Then the implicit bias of gradient descent on the second layer is characterized for a 1-dimensional regression problem, which can also be generalized to the high-dimensional case.

I think it is nice to have an explicit characterization of the minimum-kernel-norm solution. Moreover, the observation that gradient descent with the standard parameterization and initialization basically only trains the second layer is also interesting.

However, the current presentation also has many limitations:
1. Theorem 1, 2 and 6 consider gradient descent on an adjusted training set. Specifically, Theorem 1 and 2 claim the existence of u and v, which are used to adjust the training set, but it seems that how to find u and v is not discussed. Moreover, above Theorem 6, it is said that "If u and v in the solution of (17) are small, then the solution is close to the solution of (16)." How should we find u and v? Can it be proved that u and v are small?
2. The function g given by Theorem 1 and 2 are similar to the results presented in (Savarese et al., 2019) and (Ongie et al., 2020). Can you include a detailed comparison with their results and proof techniques?
3. In Theorem 4, it is assumed that inf_n \lambda_\min (\Theta_n) > 0. However, this can usually be proved in the NTK setting, for example in (Simon S. Du, Xiyu Zhai, Barnabas Poczos, Aarti Singh. Gradient Descent Provably Optimizes Over-parameterized Neural Networks). Can this assumption be proved?
4. The appendices should be included.

---

> ### Author Response · Authors · 2020-11-14
> **Thanks for your feedback - All comments are addressed below**
>
> Thank you for your valuable feedback. There are a number of interesting details that we could not fit in the main part, but discussed in the Appendix. Some of your questions are addressed there. Here is our response to your comments:
>
> 1.    This is a good observation. The solution to the variational problem is independent of added linear functions, which naturally leads to the necessity to adjust the training data by a linear function. There are two reasons why we don't consider this a serious limitation. First, a linear function does not make a big difference in terms of the complexity, and in our result we care more about the curvature. Second, in experiments we found that the adjustment of the data was not needed in order for the theoretical description to be accurate.
>
>    Finding the explicit form of u and v is indeed a nontrivial problem and in general we can't guarantee that u and v are small. However, in experiments, including those shown in Figure 1, the adjustment of the training data was not necessary for the description to be accurate. Even without adjusting the training data, we found that the output of the network was close to the solution of the variational problem given in Theorem 1. This indicates that u and v are indeed small, at least in numerous experiments we conducted. In order to better understand the role of u and v, we also conducted experiments with extremely tilted data which are shown in Figure A5. Even in this case, the difference was relatively small in relation to the magnitude of the outputs of the network, barely noticeable, and only visible when we subtracted the tilt.
>
> 2.    In (Savarese et al., 2019) and (Ongie et al., 2020), they considered the networks which minimizes the 2-norm of all parameters while perfectly fitting all training data. In our work we discuss the network after training by gradient descent. So our goal is different from theirs and the result is also different. The results of (Savarese et al., 2019) and (Ongie et al., 2020) minimize the 1-norm of second derivative and 1-norm of Radon transform, while ours minimizes the 2-norm of second derivative and 2-norm of Radon transform. In addition, in our result, we have the function zeta which regulates the curvature of the function g differently at different locations. The function zeta depends on how we initialize the network's parameters. In (Savarese et al., 2019) and (Ongie et al., 2020), their results do not involve this type of location dependent function.
>
>    In relation to the proof techniques: since the problem setup is different, our calculations are different. A similarity is that, both calculations exploit the property that the second derivative of the ReLU is a delta function, which allows us to relate parameters and functions. Some details about the comparison with their results are given in Appendix C.4.
>
> 3. You are correct, this can be proved. We discussed this in Appendix C.1 where also citations are included. The paper that you mentioned is already cited.
>
> 4. The appendices are included with the submission and can be found in the OpenReview forum as a Supplementary Material zip file.

---

> > ### Comment · AnonReviewer5 · 2020-11-14
> > **Why is it natural to study the implicit bias of gradient descent on an adjusted training set?**
> >
> > Thanks for the response. I am sorry that I did not notice the supplementary file at the beginning.
> >
> > After reading the authors' response, I am still concerned about the adjusted training set. Theorem 1 and 2 only study the implicit bias of gradient descent on an *adjusted* training set, not on the original one. Moreover, this adjusted training set seems to be *unknown* (since the theorems say "there exists u and v"), and as the authors mention, in general it cannot be guaranteed that u and v are small. Is there anything we could say about the implicit bias on the original training set?
> >
> > If I understand it correctly, in Figure 1 and A5, only 1-dimensional inputs are considered. I think the 1-dimensional case is too special, and I am not convinced that we can directly generalize these empirical results to the high-dimensional case, and moreover to interesting datasets such as MNIST.
> >
> > I find the minimum-kernel-norm characterization from (Zhang et al., 2019) more interpretable, and it does not need an adjusted training set, and it can handle the high-dimensional case directly. Even though the authors claim their results are equivalent to the minimum-kernel-norm result, to make the comparison they also adjust the training set for the minimum-kernel-norm interpretation (cf. Eq. A121); however the minimum-norm result actually holds on the original training set.

---

> > > ### Author Response · Authors · 2020-11-19
> > > **Why the adjustment and why our result give a close description of training with the original data**
> > >
> > > In response to your question on how much could be said about the original problem:
> > >
> > > Here we briefly explain why the solution of training on the original data can be described approximately by the solution to the variational problem in our Theorem 1. The reason is that fitting a linear function requires a far smaller change of the parameters than fitting a general function. So, if we fit the linearly adjusted data, we only need to adjust the parameters a little bit in order to fit the original data. Since the change of parameters is so small, the solution to the variational problems (15) and (17) are very close, which means that the variational problem in Theorem 1 also gives a close description of training with the original data. We plan to add a detailed discussion in the appendix.
> > >
> > > As mentioned earlier, this is also what we have observed in numerous computer experiments. We found that the description of the implicit bias given in our theorem was very accurate even if one does not adjust the data. To provide further evidence of this, we are conducting additional experiments.
> > >
> > >
> > > In response to your comment about our results in relation to the work of Zhang et al.:
> > >
> > > Upon linearly adjusting the training data we obtain an explicit description of the bias introduced by gradient descent training, related to cubic splines. As mentioned above, our results also give a close approximation of training on the original data. Our results describe precisely the role of the parameter initialization distribution and how it determines the inverse curvature penalty function $\zeta$. This gives us a rather good picture of how the initialization scheme affects the implicit bias of gradient descent. Also our result allows us to relate early stopping to smoothing splines. Although the minimum-kernel-norm characterization of Zhang et al. (2019) can describe training on the original data accurately, their results do not give us any of the above insights.
> > >
> > > To provide some further context about the linear adjustment of the training data, we can point at important previous works where similar adjustments have also been used. Ongie et al. (2020) also used a linear adjustment in order to derive an explicit type of characterization in their setting.
> > > - The object $\bar R_1(f)$ that they characterize in their Theorem 1 involves a linear adjustment of the function f which has no explicit description. The corresponding notion without linear adjustment is $\bar R(f)$, for which an estimate is given.
> > > - The R-norm in their Definition 1 is strictly speaking a semi-norm, and the R-norm of a function f is independent of linear functions added to $f$. So their result is also incapable of handling linear function and that's the reason why they also need to do linear adjustment.

---

### Author Response · Authors · 2020-11-24
**General response to Reviewers**

We would like to thank all reviewers for their helpful comments. They are very important to improve our paper. We have already revised and uploaded the main text and the supplementary materials according to your suggestions. Here is the summary of modifications:

1. We moved the Section of notations and problem setup before the statement of our main result to improve the readability of the main theorem.

2. We added the Appendix L to explain why our Theorem 1 also approximately describe the gradient descent training with original training data. The high level intuition is that fitting a linear function only requires a very small adjustment of the parameters of the network in comparison with the parameter adjustment needed to fit a non-linear function.

3. We added a paragraph in Appendix C.4 to give more details about the difference between our result and the result of Savarese et al. (2019). We pointed out that our characterization gives a RKHS norm while Savarese et al. (2019) gives a non-RKHS norm. We explained why there is such a difference.

4. We added the Remark A3 in the Appendix E and discussed that in which case the rank assumption is satisfied. We also mentioned the rank assumption in the main text.

5. We fixed the typos, the notation clashes and unclear statements pointed out by the reviewer. For example, we used $\phi$ for activation functions and $\sigma$ for the standard deviation of Gaussian distributions.

6. We added the conclusion part at the end of the main text. We summarized the main result and discussed the possible extensions. We showed that our result may help to explain the generalization performance of the networks trained by gradient descent. We explained how our result can give insight on how to choose the initialization scheme to achieve better generalization, which is of interest to some reviewers. At the end, we related our result to early stopping and the spectral bias.

---

### Decision · Program_Chairs · 2021-01-07
**Final Decision**

**Decision:**

Reject

**Comment:**

This paper analyzes the implicit bias of gradient descent of infinite width 2-layer neural networks with ReLU activation. It is shown that the dynamics of gradient descent to optimize the 2-layer NN converges to the optimization dynamics on the random feature model in the infinite width limit. Then, it is shown that the gradient descent converges the minimal L2 norm solution from the initial parameters which yields regularization on a weighted integration of second order differentiation. Although this type of analysis has been given in the existing work, this paper gives its explicit form in 1-dimension input setting.

This paper reveals an interesting fact about the implicit regularization that would be educationally valuable. On the other hand, I should mention that there is room for improvement in its theoretical contribution and moreover its novelty is rather limited.
1. Although the explicit formulation of the implicit regularization is informative, the minimum norm bias itself is already pointed out by existing work and this work follows the line. Especially, regularization on the second order derivative has been already pointed out by previous work (although they are 1-norm regularization).
2. The logical jump from the original data to the adjusted data is still not convincing. It is explained that some numerical experiments show the linear term is negligibly small, which means the problems (15) and (17) are very close. However, this excuse does not make sense for this kind of "theoretical" work. The logic used here should be clarified to make the theoretical framework complete.

The evaluations by the reviewers indicate that this paper is on the borderline, and I also feel that some more additional strong point would be required so that this paper is accepted. I encourage the authors to go in this direction and make the analysis more detailed so that the theoretical framework would get more completed.

Minor comment:
Theorem 4 overlaps the result given by the following paper [R1]. It is recommended that the relation and novelty compared with that paper is discussed.

[R1] E, W., Ma, C. & Wu, L. A comparative analysis of optimization and generalization properties of two-layer neural network and random feature models under gradient descent dynamics. Sci. China Math. 63, 1235–1258 (2020).